# Few-shot Learning by Focusing on Differences

## Abstract

Few-shot classification may involve differentiating data that belongs to a different level of labels granularity. Compounded by the fact that the number of available labeled examples are scarce in the novel classification set, relying solely on the loss function to implicitly guide the classifier to separate data based on its label might not be enough; few-shot classifier needs to be very biased to perform well. In this paper, we propose a model that incorporates a simple inductive bias: focusing on differences by building a *dissimilar* set of class representations. The model treats a class representation as a vector and removes its component that is shared among closely related class representatives. It does so through the combination of learned attention and vector orthogonalization. Our model works well on our newly introduced dataset − CIFAR-Hard − that contains different levels of labels granularity. It also substantially improved the performance on fine-grained classification dataset, CUB; whereas staying competitive on standard benchmarks such as mini-Imagenet, Omniglot, and few-shot dataset derived from CIFAR.

## 1 Introduction

Progress in artificial intelligence (AI) has been rapid. AI agents have been outperforming humans in an increasing variety of tasks, such as in recognizing images on ImageNet (He et al., 2016) and in the ancient game of Go (Silver et al., 2016). However, challenges remain – systems that outperform humans usually require learning from very large-scale data. In contrast, humans only require few examples to be able to rapidly adapt to a novel task; humans are still better *learners*. Few-shot learning methods – which learn classes from few labeled examples – aim to bridge this gap.

In learning a classification algorithm from a few labeled examples, one may train the algorithm with a different set of abundantly labeled data (base set); before adapting it to the unknown examples. However, it might be the case that the available labeled examples are of different granularity level. For example, it is possible that it is only trained to differentiate between cats and dogs, but is tested on differentiating different breeds of dogs. The set of labeled examples is also very limited; in the extreme case, only one labeled example is provided for each class (called one-shot classification). A few-shot learning algorithm can learn to identify features that are important for doing well on the base set – these can be adapted to classify the few labeled examples as long as the domain remains similar. But with so few examples provided and possible differences in the task granularity, few-shot learning algorithms need to be very biased to perform well. The question is then: what kind of bias is reasonable?

**Our work.** In this paper, we propose a model that performs classification in a novel task by focusing on the differences between closely related classes of its support set. Our bias is loosely inspired by how scientists often work (Mill's method of difference): in looking for potential causes of a phenomenon, a scientist would often focus on the differences in the circumstances (features) in the instance in which the phenomenon occurred and the circumstances in instances for which the phenomenon did not occur (Mill, 1875). Our method focuses on the differences by removing components in each class representative that are shared with closely related classes.

**Our contributions.**

1. We introduce a method which incorporates a simple yet effective inductive bias: focusing on differences, and show that it works well on classifying closely related classes. Our

results show that the method achieves better performance on a standard fine-grained classification benchmark (CUB dataset) and on our proposed benchmark consisting of a mix of fine and coarse-grained classification tasks, CIFAR-Hard. On other commonly used benchmark datasets, CIFAR-FS, *mini*-ImageNet, and Omniglot, its performance is competitive with existing methods.

2. We propose a methodology to build harder few-shot learning datasets without requiring very large hierarchically labeled datasets. Using this methodology, we build CIFAR-Hard – a mix of fine-grained and coarse-grained few-shot classification dataset derived from CIFAR-100. Our empirical evaluation shows that the currently existing methods are not well equipped to handle this scenario.

## 2 DISSIMILARITY NETWORK

### 2.1 FEW-SHOT LEARNING

In few-shot learning, we are given a *base set* $\mathbb{B}$ and a *novel set* $\mathbb{N}$. The base set contains labeled examples from a large number of classes while novel set contains classes not found in the base set. The objective of few-shot learning is to train a classification algorithm $P$ on the training set $\mathbb{X}_{train} = \mathbb{B}$, in such a way that it generalizes to the elements of the novel set $\mathbb{N}$. Some methods may also train on the small number of labeled examples from the novel set. In that case, the training set becomes $\mathbb{X}_{train} = \mathbb{B} \cup \mathbb{N}_{labeled}$, with labeled novel set $\mathbb{N}_{labeled} \subset \mathbb{N}$. In one-shot learning, the novel set only contains one labeled example for each class, while for $k$-shot learning, the novel set contains $k$ examples for each class.

We use the *episodic training* proposed by Vinyals et al. (2016) to make sure that the training and test condition match. At every step, we sample some examples to form an episode $\mathbb{T} \subset \mathbb{X}_{train}$ to train the classification algorithm $P$. Each episode $\mathbb{T}$ consists of a small set of $N$-labeled examples (called support set) $\mathbb{S} = \{(\boldsymbol{x}_i, y_i)\}_{i=1}^{N}$ and a set of $M$ examples to be labeled (called query set) $\mathbb{Q} = \{\boldsymbol{x}_i\}_{i=1}^{M}$, where $\boldsymbol{x}_i \in \mathbb{R}^D$ is a $D$-dimensional feature vector with label $y_i$ (for computing the loss on the query set), simulating the conditions for learning the novel set $\mathbb{N}$. We denote the set of labeled examples from the support set with class $k \in \{1, ..., K\}$ in an episode as $\mathbb{S}_k$.

Our method adopts the approach of similarity learning. Instead of learning a distance or similarity function, we learn a *space* (embedding) that works well with a fixed similarity-based classifier in that space. Specifically, our model learns to construct a space that is optimized to separate data that belong to different classes for a classifier that uses dot-product as its similarity function. We define two levels of embedding based on how the embedding utilizes task information:

**Global task embedding** learns an embedding function that is optimized for all episodes that it is trained upon. The assumption is that the produced embedding will learn a meaningful and general representation that is sufficient to separate data points on the novel task without knowing what classes appear in it.

**Task-aware embedding** removes the assumption that the learned global embedding is sufficient for the novel task. Instead, it takes the possible classes of the novel task into account. On the novel task, it will embed the query set conditioned on the support set that is aware of its member, giving the full context to the prediction. Our model builds an *explicit task-aware embedding* that separate a class from the weighted average of its closely-related classes. It is explicit in the sense that our model explicitly encodes such an inductive bias into its function (architecture). In contrast, *implicit task-aware embedding* only relies upon its loss function to adjust its function as to induce separation between classes.

### 2.2 MODEL

For each class, our model – which we call the Dissimilarity Network – computes a class representative, called a *prototype*, that is used to classify a new instance of data by comparing its similarity to the prototype, through an inner product. For 1-shot classification, each training example in the novel task is transformed into the prototype, while for $k$-shot classification, the mean of the $k$ instance representations that belong to each class is transformed into the prototype.

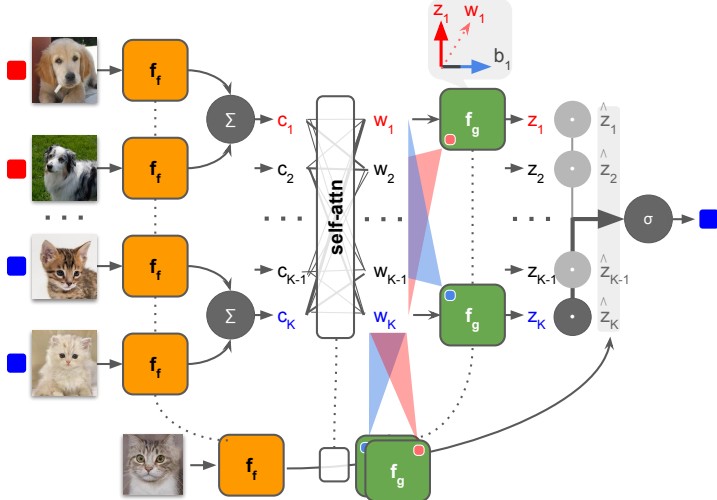

Figure 1: Dissimilarity Network architecture

The Dissimilarity Network computes prototypes that are *dissimilar* to one another, in the sense that the components of the class prototypes are orthogonal to the direction of the weighted average of closely-related classes. This lifts the burden of the classifier as the classes are more easily distinguishable – since what remains are their differences. The model works by iteratively enhancing the representation of the prototypes through a set of learned transformations (embedding functions). The first transformation builds a global task embedding through a learned dimensionality-reduction function. The global embedding retains features that are useful for the training tasks but does not take into account the classes that are present in the novel task. The second transformation transforms the prototypes into a task-aware embedding using self-attention networks, taking into account other classes that are present in the task. Finally, the last transformation computes the class prototypes that are dissimilar, by locally orthogonalizing the representations to the weighted average of other closely related class representations.

When presented with a new point to classify, it computes the global embedding for that point, transform it using the task-aware embedding, locally orthogonalize the point for each possible class, and select the most similar prototype's class as the class of the new point. Figure 1 illustrates how the model constructs class representations as well as predict the new point.

### 2.2.1 GLOBAL EMBEDDING

We learn a feature extraction function $f_f : \mathbb{R}^D \to \mathbb{R}^H$ for the images to reduce their representation to a $H$-dimensional vector. We used deep convolutional neural networks (Krizhevsky et al., 2012) which captures the local interaction of neighboring pixels and builds a hierarchical representation of them. This feature extractor constructs our first level of embedding. It is trained to extract information that is useful for all the training episodes but is not specialized to a particular novel task. We use the global embedding to compute the class mean (or prototype) $c_k$ of the $H$-dimensional representation of the support points,

$$c_k = \frac{1}{|\mathbb{S}_k|} \sum_{(\boldsymbol{x}_i, y_i) \in \mathbb{S}_k} f_f(\boldsymbol{x}_i). \tag{1}$$

### 2.2.2 SELF-ATTENTION

By seeing the prototypes as a set of vector: $\mathbb{C} = \{c_i\}_{i=1}^K$, we can learn a set-to-set function that conditions the member of the set to every other member within the set. This gives *task-awareness* to our prototypes; each member is now aware of the other class representatives of the given task.

Our set-to-set operation is based on the self-attention mechanism introduced by Vaswani et al. (2017). Given a query, an attention mechanism learns to "attend" – by means of weighted aver-

age – to different parts of the set depending on how relevant it is to the query. In self-attention, the query is the member of the set itself. Our self attention uses embedding functions for query $h_Q$, key $h_K$, and value $h_V$. Each embedding function is parameterized by a neural network that computes a mapping $\mathbb{R}^{K \times H} \to \mathbb{R}^{K \times H}$. For simplicity we assume that for any input in the form of a set of vector $\mathbb{A}$, it will automatically be cast into matrix $\boldsymbol{A} \in \mathbb{R}^{K \times H}$, whereas the output will be cast back into a set. The self-attention mechanism $self\text{-}attn : \mathbb{R}^{K \times H} \to \mathbb{R}^{K \times H}$ is formulated as follows.

$$self\text{-}attn(\boldsymbol{C}) = softmax\Big(\frac{h_Q(\boldsymbol{C})h_K(\boldsymbol{C})^T}{\sqrt{H}}\Big)h_V(\boldsymbol{C}) \tag{2}$$

Intuitively, the self-attention computes a weighted average of the element of the input matrix $\boldsymbol{C} \in \mathbb{R}^{K \times H}$, representing a set of prototypes $\mathbb{C}$. This operation has the effect of averaging out noisy components from global embedding that may be relevant for other tasks but are irrelevant to the set of classes in the current novel task. The weights are obtained using the learned attention function, which in this case, is parameterized by $h_Q$ and $h_K$. In our case, our prototype $\boldsymbol{c}_k$ learns to be aware of the other class representatives $\mathbb{C} \setminus \boldsymbol{c}_k$ by incorporating some of their components.

We use bidirectional LSTM (BLSTM) (Hochreiter & Schmidhuber, 1997) for our attention embedding function for query $h_Q$ and key $h_K$, while using either identity function or BLSTM on $h_V$. BLSM computes a concatenation of two sequence of opposing-direction by sequentially applying the element $\boldsymbol{x}_t \in \mathbb{R}^H$ of its input $\boldsymbol{x} = [\boldsymbol{x}_1, ..., \boldsymbol{x}_T]$ into an LSTM, $\boldsymbol{h}_t, \boldsymbol{u}_t = LSTM(\boldsymbol{x}_{t-1}, \boldsymbol{h}_{t-1}, \boldsymbol{u}_{t-1})$. The computation yields a sequence of vector that each are conditioned on its neighboring elements. Our BLSTM uses context sharing between key attention embedding $h_K$ and query attention embedding $h_Q$.

We are aware of the sequential nature of BLSTM, which can be counter-intuitive as we are modeling a set-to-set operation which should not have any preference for ordering. However, we found empirically that this setup offers more performance gain compared to the use of traditional linear function as attention embedding function. The BLSTM may learn to ignore the unimportance of set ordering due to the nature of episodic training, which exposes it to many permutations of the possible class-orderings. Moreover, the attention also gives a global context of the member of the set, which could further alleviate the ordering issues (if any).

### 2.2.3 FOCUSING ON DIFFERENCES

We encode our inductive bias in the form of neural network architecture, in which we remove the components that are shared among other class representatives, thereby giving the model the ability to *focus* only on the inter-class differences. Since we treat the prototypes as a vector, one natural way to achieve that is by making the prototypes to be locally orthogonal to the components that are shared among other classes. We learn to find such components by using dot-product attention.

For a $H$-dimensional prototype $\boldsymbol{c}_k \in \mathbb{C}$ of class $k$, it will have a corresponding task-aware vector prototype $\boldsymbol{w}_k \in \mathbb{W} = self\text{-}attn(\boldsymbol{C})$ following the method described in the Section 2.2.2. Let $\mathbb{W}' = \mathbb{W} \setminus \boldsymbol{w}_k$, the components shared among the other classes $k' \neq k$ that are locally orthogonal to the vector prototype $\boldsymbol{w}_k$ is computed using attention function $attn : \mathbb{R}^H \times \mathbb{R}^{K \times H} \to \mathbb{R}^H$:

$$attn(\boldsymbol{w}_k, \boldsymbol{W}') = softmax\Big(\frac{\boldsymbol{w}_k \cdot \boldsymbol{W}'^T}{\sqrt{H}}\Big)\boldsymbol{W}' \tag{3}$$

Essentially, through weighted averaging, it selects components from the other class prototypes based on how similar the embedded vector prototype $\boldsymbol{w}_k$ is to them.

We make the prototype $\boldsymbol{w}_k$ to be orthogonal to the shared components of $\mathbb{W} \setminus \boldsymbol{w}_k$ by removing its projection to the shared components. Specifically, we use the shared components $\boldsymbol{b}_k = attn(\boldsymbol{w}_k, \mathbb{W} \setminus \boldsymbol{w}_k)$ that belongs to class $k$ as the basis of the projection $proj(\boldsymbol{w}_k, \boldsymbol{b}_k)$ as follows:

$$proj(\boldsymbol{w}_k, \boldsymbol{b}_k) = \frac{\boldsymbol{w}_k \cdot \boldsymbol{b}_k}{||\boldsymbol{b}_k||^2}\boldsymbol{b}_k \tag{4}$$

Intuitively, it *maps* the vector of prototype $\boldsymbol{w}_k$ to represent its direction using the given basis $\boldsymbol{b}_k$ (i.e., projecting it into the basis). It produces components that are linearly dependent on the basis, thereby removing it: $\boldsymbol{z}_k = \boldsymbol{w}_k - proj(\boldsymbol{w}_k, \boldsymbol{b}_k)$ will produce a new prototype representation $\boldsymbol{z}_k$ that is orthogonal to the weighted average of the components of other closely-related prototypes.

### 2.2.4 Classification and Learning

To classify a new point $\hat{\boldsymbol{x}}$, we follow similar transformations as computing the locally orthogonal prototypes. First global embedding is computed for the point $\hat{\boldsymbol{v}} = f_f(\hat{\boldsymbol{x}})$. For consistency, we follow the transformation given by $self\text{-}attn$. However, since we only have a single point $\hat{\boldsymbol{v}}$, we construct a set of length $K$ by duplicating the point $K$ times as the input: $\hat{\mathbb{W}} = self\text{-}attn(\{\hat{\boldsymbol{v}}\}_{i=1}^K)$.

If the attention value embedding $h_V$ (Section 2.2.2) is an identity function (or any other element-wise function), any $\forall_{\hat{\boldsymbol{w}}_k \in \hat{\mathbb{W}}} \hat{\boldsymbol{w}}_k = \hat{\boldsymbol{v}}$ since its taking a weighted average of a set of identical vectors. When $h_V$ is a BLTSM (or function that operates on a set of elements), $h_V$ transforms the vector through a multi-stage non-linear processing.

For a task-aware embedding $\hat{\boldsymbol{w}}_k \in \hat{\mathbb{W}}$, we compute the vector that is locally orthogonal to the set of prototypes of class $\tilde{k} \neq k$ by computing $\hat{\boldsymbol{z}}_k = \hat{\boldsymbol{w}}_k - proj(\hat{\boldsymbol{w}}_k, \boldsymbol{b}_k)$ with basis given by $\boldsymbol{b}_k = attn(\boldsymbol{w}_k, \mathbb{W} \setminus \boldsymbol{w}_k)$ from the Section 2.2.3.

Given an unlabeled data $\hat{\boldsymbol{x}}$, the transformations gives a set of locally orthogonalized vector $\{\hat{\boldsymbol{z}}_i\}_{i=1}^K$ for comparison with the locally orthogonal prototypes $\{\boldsymbol{z}_i\}_{i=1}^K$. Dissimilarity Network then computes a distribution over classes for point $\hat{\boldsymbol{x}}$ using Softmax over inner product:

$$p(y = k|\hat{\boldsymbol{x}}) = \frac{\exp(\langle \hat{\boldsymbol{z}}_k, \boldsymbol{z}_k \rangle)}{\sum_{k'} \exp(\langle \hat{\boldsymbol{z}}_{k'}, \boldsymbol{z}_{k'} \rangle)} \tag{5}$$

Learning is done using the cross-entropy loss with the label of the instance.

## 3 Related Works

There are many works on few-shot learning, which was started on the assumption that currently, learned tasks can help in making a prediction in a new task (Fei-Fei et al., 2006). It soon gained interest from many researchers, which introduced many interesting techniques that contribute to huge strides of progress in few-shot learning. We will quickly review some of the recently proposed methods and delve deeper into metric learning-based methods that are more related to our work.

**Transfer learning-based methods** follows the standard transfer learning procedure (network pre-training & fine-tuning). Gidaris & Komodakis (2018); Qi et al. (2018); Chen et al. (2019) propose to directly predicting weights of the classifiers on a novel task.

**Initialization based methods** address few-shot learning by finding a way to better initialize a model. Ravi & Larochelle (2016) uses LSTM as a meta-optimizer to rapidly adapt neural network on the novel task, whereas Munkhdalai & Yu (2017) uses external memory to update weights. Another line of works is concerned about finding a good initialization, as such that finetuning can be done using fewer steps (Finn et al., 2017; Nichol & Schulman, 2018; Rusu et al., 2018).

**Metric and similarity learning-based methods** assumes that representation produced by some model on a task share some similarities with those that are produced by another task. Essentially, the goal is to learn a comparison model that can distinguish different classes on the novel task by measuring its distance or similarity to some representation produced by the support set.

Our proposed method is similar to the prototypical networks (Snell et al., 2017) – and subsequently Mensink et al. (2013) – in its use of mean representation of class (or prototypes). The similarity stops there, as the prototypical networks directly perform classification by comparing the distance of the new input to each prototype. They assume that the embedding function that produces the prototypes can sufficiently capture useful and general enough representation that is transferable to the novel set; it only computes global task embedding. As shown in our results, their assumption breaks down when there are changes in class granularity or the label is of fine granularity. In contrast, our method does not directly classify on the prototypes; instead, it transforms the prototypes, producing task-aware embeddings that are locally orthogonal to the shared components belonging to different classes. Thus, classification is performed by computing a Softmax over dot product between the new point and the task-aware prototypes.

The Dissimilarity Network uses context embedding similar to the full context embedding (FCE) extension of the matching network (Vinyals et al., 2016). However, there are some glaring differences

in how they operate. The matching network carries the entire support for the prediction. It predicts the label of an unknown point by computing the linear combination of the label of its support set; as the support set grows, the memory increases linearly with it. Their full context embedding conditions the prediction on the entire support set; computing the task-aware embedding is quadratic in the number of elements of the support set. Moreover, as they do not construct an explicit reference for the classes to condition into, it is less clear how reasonable separation of a point belonging to different classes can be maximized. As pointed out by Snell et al. (2017), the FCE extension of the matching network does not make that much difference.

Our model only maintains a set of prototypes and classifies a new point based on how orthogonal its representation to the prototypes. Since we condition the prediction only on the prototypes, it will not grow with the size of the support set. Moreover, computing the task-aware embedding is only quadratic in the number of prototypes (i.e., labels) – as opposed to the number of support set. We also explicitly computes the representations to be dissimilar; lifting the reliance on learning sufficiently separable inter-class representations only to the loss function.

Our similarity function can be set differently. One possible extension is to use learned similarity or distance function similar to RelationNet (Sung et al., 2018), which we leave for future work.

## 4  EXPERIMENTS

**Datasets & scenarios.** We evaluate all models on the standard dataset that is widely used in few-shot learning: omniglot, miniImageNet, and CUB. Apart from that, we also evaluate all models on CIFAR dataset with the two splits: hard and normal split (which we will elaborate later).

**Evaluation.** Our evaluation follows Chen et al. (2019), that is by sampling $N$-class to form $N$-way classification (with N=5 unless otherwise stated). For $k$-shot task, we pick $k$ labeled instances for each class to be the support set and 16 instances for query set. All results are averaged over 600 experiments which follow the above settings. We evaluate all models on 1-shot and 5-shot setting, which is the most common setting adopted in few-shot learning.

**Implementation details.** All methods are trained using Adam optimizer Kingma & Ba (2014) with the initial learning rate of $10^{-3}$, which we cut half every 2000 episodes. We apply the following standard data augmentation on all datasets (except CIFAR): random crop, right-left flip, and color jittering. Following Snell et al. (2017), we use a four-layer convolution backbone (Conv-4) with an input size of 84x84 as a feature extractor for all methods. We use the open-source implementation of Chen et al. (2019) for other methods that we reported. We pick the best performing model based on the validation for meta-learning methods, whereas for baseline and baseline++ Chen et al. (2019), we follow the recommended settings prescribed in their paper. We trained our model on 800 epochs. Our best performing model on all datasets, based on validation set, uses identity function for the value attention embedding $h_V$ except on the CUB dataset, which uses BLSTM.

### 4.1  STANDARD BENCHMARKS

**Omniglot** (Lake et al., 2011) dataset consist of 1623 handwritten digits from 50 different alphabets. There are 20 examples per character which is drawn by different people. We follow Vinyals et al. (2016) procedure for evaluation.

*mini*-**ImageNet** is a 100 classes subset of ImageNet (Deng et al., 2009) dataset (ILSVRC-12 dataset), which was first proposed by Vinyals et al. (2016). It consists of 600 images per class. Recent works follow the setting proposed by Ravi & Larochelle (2016), which consist of randomly selected 64 base, 16 validation, and 20 novel classes.

**CUB** or Caltech-UCSD Birds 200-2011 dataset (Wah et al., 2011) is a fine-grained classification dataset which consist of 200 classes (bird species) and 11,788 images in total. We follow Hilliard et al. (2018) setting which is composed of 100 base, 50 validation, and 50 novel classes.

**CIFAR-FS** dataset is derived from CIFAR dataset (Krizhevsky et al., 2009) consist of 60,000 32x32 color images with 100 classes (belonging to 20 superclasses), with 600 images for each class. Our split consists of randomly sampled 40 base, 15 validation, and 45 novel classes (detail on appendix).

## 4.2 HARDER BENCHMARKS

**General setting.** Our approach requires the dataset that we derived from to have at least two different levels of class granularity. For example, CIFAR dataset which has two levels of labels granularity. ImageNet labels also form a hierarchy which – through this method – can be derived into several hard few-shot classification datasets. In the case where different labels of granularity are absent, one may be able to construct new labels by exploring the natural hierarchy which may present.

**Method.** Given a J-labeled dataset $\mathbb{D} = \{(\boldsymbol{x}_1, y_1^{coarse}, y_1^{fine}), ..., (\boldsymbol{x}_J, y_J^{coarse}, y_J^{fine})\}$ where the labels comes from a two-level hierarchy: coarse-grained label $y_i^{coarse} \in \mathbb{K}^{coarse}$ and fine-grained label $y_i^{fine} \in \mathbb{K}^{fine}$. $\mathbb{K}_s^{fine}$ denote a subset of labels $\mathbb{K}^{fine}$ that belongs to coarse-grained label (superclass) $s \in \mathbb{K}^{coarse}$.

For all coarse-grained label $y_i^{coarse} \in \mathbb{K}^{coarse}$, select some $y_i^{fine} \in \mathbb{K}^{fine}$ that is the subclasses of $y_i^{coarse}$ (i.e., $\mathbb{K}_{y_i^{coarse}}^{fine}$), producing a set of fine-grained labels from all superclasses $\mathbb{K}_{base}^{fine}$. Construct the base set: $\mathbb{B} = \{(\boldsymbol{x}_i, y_i^{coarse}) | (\boldsymbol{x}_i, y_i^{coarse}, y_i^{fine}) \in \mathbb{D}, y_i^{fine} \in \mathbb{K}_{base}^{fine}\}$. The novel set can be built by taking the rest of unused data and pair them with their fine-grained labels: $\mathbb{N} = \{(\boldsymbol{x}_i, y_i^{fine}) | (\boldsymbol{x}_i, y_i^{coarse}, y_i^{fine}) \in \mathbb{D}, y_i^{fine} \notin \mathbb{K}_{base}^{fine}\}$. Validation set is constructed the same way as novel set – we leave out the detail of its construction for simplicity.

This approach (illustrated in Figure 2) is advantageous as on each task, the labels can vary from being fine-grained to coarse-grained depending on the random selection. As such, the methods that rely on the awareness of the overall novel tasks will likely to fail as it builds a general embedding that works on all but not optimized for the current task. On the other hand, a dynamic method that conditions the prediction on the support set of the given task will likely to perform better.

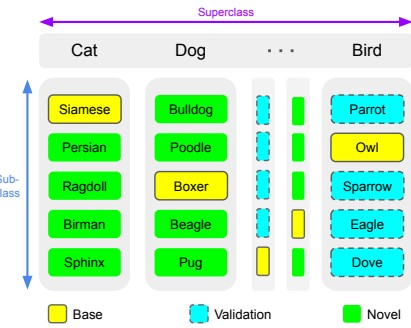

Figure 2: Illustration on construction of harder dataset. The base dataset uses the superclass taxonomy (e.g., cat, dog, and bird) instead its subclass (e.g., siamese, boxer, and owl).

**CIFAR-Hard** is derived from CIFAR-100, using the aforementioned method. We derive our harder benchmark from CIFAR-100 because it has two different level of labels granularity. The size of the dataset is also not too big, making it suitable for use as a benchmark. The following is the detail for each split (more detail on the appendix). 20 coarse-grained base classes from 40 fine-grained classes (derived from the entire 20 superclasses, 2 classes each). 15 validation classes (derived from 5 superclass, 3 classes each). 45 novel classes (derived from 15 superclass, 5 classes each).

## 4.3 RESULTS & DISCUSSIONS

Table 1 shows how our method fares against others. Our method performs the best on a fine-grained classification task such as CUB and improved on a wide margin on its 1-shot classification task. As we have suspected, there is an increasing need to focus on the difference between classes when the classification task becomes increasingly fine-grained. Despite also being trained on fine-grained classification tasks on the CUB dataset, our inductive bias still seems to be helpful in classifying similar-looking classes as it further separate class representation by explicitly removing latent features shared among those classes. At a glance, it appears as if the 1-shot performance improvement our method is significantly higher than its improvement in 5-shot. However, this is due to the fact that the Baselinee++ uses retraining, which relies heavily on the availability of labeled data on the novel

Table 1: 5-shot and 1-shot classification accuracy on all datasets. Four-layer convolution backbone (Conv-4) with an input size of 84x84 is used as a feature extractor for all methods for a fair comparison. No finetuning is performed on CIFAR-H and CIFAR-FS, all the hyperparameters are the one used in the *mini*-ImageNet dataset. All accuracy results are averaged over 600 test episodes and are reported with 95% confidence intervals. *Results reported by Chen et al. (2019).

| | 5-shot classification accuracy (%) | | | | | |
| --- | --- | --- | --- | --- | --- | --- |
| Method | CIFAR-H | CUB | CIFAR-FS | *mini*-ImageNet | Omniglot | Average |
| DissimilarityNet (Ours) | $68.45 \pm 0.74$ | $81.23 \pm 0.63$ | $71.10 \pm 0.71$ | $65.40 \pm 0.61$ | $99.27 \pm 0.10$ | $77.09 \pm 0.56$ |
| MatchingNet (Vinyals et al., 2016) | $63.34 \pm 0.78$ | $72.86 \pm 0.70^*$ | $67.14 \pm 0.77$ | $63.48 \pm 0.66^*$ | $99.37 \pm 0.11^*$ | $73.24 \pm 0.60$ |
| ProtoNet (Snell et al., 2017) | $64.30 \pm 0.81$ | $70.77 \pm 0.69^*$ | $69.96 \pm 0.77$ | $64.24 \pm 0.72^*$ | $99.15 \pm 0.12^*$ | $73.68 \pm 0.62$ |
| MAML (Finn et al., 2017) | $62.87 \pm 0.77$ | $72.09 \pm 0.76^*$ | $65.98 \pm 0.81$ | $62.71 \pm 0.71^*$ | $99.53 \pm 0.08^*$ | $72.64 \pm 0.63$ |
| RelationNet (Sung et al., 2018) | $63.15 \pm 0.83$ | $76.11 \pm 0.69^*$ | $68.87 \pm 0.76$ | $66.60 \pm 0.69^*$ | $99.30 \pm 0.10^*$ | $74.81 \pm 0.61$ |
| Baseline++ (Chen et al., 2019) | $57.25 \pm 0.77$ | $79.34 \pm 0.61^*$ | $59.86 \pm 0.80$ | $66.43 \pm 0.63^*$ | $99.38 \pm 0.10^*$ | $72.45 \pm 0.58$ |

| | 1-shot classification accuracy (%) | | | | | |
| --- | --- | --- | --- | --- | --- | --- |
| Method | CIFAR-H | CUB | CIFAR-FS | *mini*-ImageNet | Omniglot | Average |
| DissimilarityNet (Ours) | $51.02 \pm 0.89$ | $65.82 \pm 0.94$ | $54.66 \pm 0.82$ | $49.34 \pm 0.78$ | $97.90 \pm 0.25$ | $63.75 \pm 0.74$ |
| MatchingNet (Vinyals et al., 2016) | $50.42 \pm 0.92$ | $61.16 \pm 0.89^*$ | $53.92 \pm 0.92$ | $48.14 \pm 0.78^*$ | $97.78 \pm 0.30^*$ | $62.28 \pm 0.76$ |
| ProtoNet (Snell et al., 2017) | $47.16 \pm 0.90$ | $51.31 \pm 0.91^*$ | $50.08 \pm 0.88$ | $44.42 \pm 0.84^*$ | $98.01 \pm 0.30^*$ | $58.20 \pm 0.77$ |
| MAML (Finn et al., 2017) | $48.64 \pm 0.93$ | $55.92 \pm 0.95^*$ | $51.78 \pm 0.94$ | $46.47 \pm 0.82^*$ | $98.57 \pm 0.19^*$ | $60.28 \pm 0.77$ |
| RelationNet (Sung et al., 2018) | $50.78 \pm 0.95$ | $62.45 \pm 0.98^*$ | $54.24 \pm 0.93$ | $49.31 \pm 0.85^*$ | $97.22 \pm 0.33^*$ | $62.80 \pm 0.81$ |
| Baseline++ (Chen et al., 2019) | $39.30 \pm 0.70$ | $60.53 \pm 0.83^*$ | $43.38 \pm 0.73$ | $48.24 \pm 0.75^*$ | $95.41 \pm 0.39^*$ | $57.37 \pm 0.68$ |

set. If we take a look at the rest of the methods which are based on meta-learning (i.e., optimization-based and metric learning), they all suffer equally on both 1-shot and 5-shot; especially on 1-shot as it is of higher variance. The difference between our method and the other methods averaged is consistent on both shots, which is around 7%.

Our method also significantly surpassed competing methods on the harder benchmark, CIFAR-H, Its improvement is slightly higher on 5-shot – compared to the 2nd best performing model. It is expected as the higher the number of support set, the more likely our method finds the most representative prototypes; as the variance of the samples for the mean decreases. Due to the nature of how the dataset is constructed, to perform well on this dataset, methods should be able to dynamically adapt its classification function based on the level of granularity presented. As our method explicitly induce dissimilar prototypes, it's able to fare significantly better compared to others. Overall, there is an average drop of 3.7% in terms of method's performance between CIFAR-H and CIFAR-FS – confirming CIFAR-H to be in fact, harder. On CIFAR-FS, the normal variant of the CIFAR dataset, our method performed slightly better on 5-shot classification. On *mini*-ImageNet, our method performed comparably with the rest, while being slightly worse on the Omniglot dataset. This is expected, as adding more inductive bias may only hurt its asymptotic performance.

To summarize, our method performed the best in its intended scenario – when the granularity of the labels are fine enough (or changing), as such that the model has to be able to dynamically adapt to it. In the case when the granularity of the labels is fixed, or when the labels are quite coarse, our method will perform comparably to the others. When accuracy is already high, our method might fail to reach optimum asymptotic performance due to the additional constraint that we impose.

## 5 CONCLUSION

We have proposed Dissimilarity Network for few-shot learning based on the idea of focusing on differences in the class representation. Our approach directly addresses the failure modes of some few-shot classifiers that do not explicitly take into account the classification task at hand, yielding non-satisfactory results on some task such as fine-grained novel classification with coarse-grained base classification task. To demonstrate the necessities of building task-aware embedding for such task, we came up with a challenging dataset, CIFAR-Hard, which we have shown to be harder than the CIFAR-FS. Dissimilarity Network introduced an architectural inductive bias which removes the shared components among classes in the prototypes by orthogonalizing them (i.e., removing their projected components) to their leave-self-out weighted local average. Our method performs comparably to the state-of-the-art methods on standard benchmarks such as Omniglot and *mini*-ImageNet, and substantially outperform other methods on CUB dataset and on the newly constructed CIFAR-Hard dataset.

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

# A  APPENDIX

## A.1  SPLITS FOR CIFAR

**Base:** forest, house, television, wolf, cloud, sweet_pepper, dinosaur, tank, caterpillar, cup, sunflower, whale, can, bottle, road, crocodile, woman, bear, otter, willow_tree, snail, aquarium_fish, girl, trout, bowl, worm, pear, streetcar, castle, flatfish, lobster, turtle, poppy, orchid, man, seal, lamp, lawn_mower, beetle, clock

**Validation:** oak_tree, kangaroo, mushroom, porcupine, squirrel, lizard, train, spider, keyboard, maple_tree, bicycle, orange, lion, rabbit, motorcycle

**Novel:** fox, boy, skyscraper, bridge, mouse, shrew, plain, possum, tiger, tulip, wardrobe, sea, couch, mountain, leopard, camel, shark, plate, dolphin, table, bee, pickup_truck, palm_tree, beaver, baby, bus, butterfly, ray, apple, cattle, crab, pine_tree, raccoon, tractor, chair, rose, telephone, chimpanzee, snake, bed, hamster, skunk, cockroach, rocket, elephant

## A.2  SPLITS FOR CIFAR-HARD

**Base:** aquatic_mammals, fish, flowers, food_containers, fruit_and_vegetables, household_electrical_devices, household_furniture, insects, large_carnivores, large_man-made_outdoor_things, large_natural_outdoor_scenes, large_omnivores_and_herbivores, medium_mammals, non-insect_invertebrates, people, reptiles, small_mammals, trees, vehicles_1, vehicles_2

**Validation:** rabbit, hamster, bed, house, kangaroo, lamp, skyscraper, squirrel, castle, table, chimpanzee, telephone, television, wardrobe, elephant

**Novel:** baby, beaver, beetle, bicycle, bottle, bus, butterfly, can, caterpillar, crocodile, cup, dolphin, flatfish, forest, girl, lion, lobster, man, mountain, oak_tree, orange, orchid, otter, pear, pickup_truck, pine_tree, porcupine, possum, ray, rocket, rose, sea, shark, skunk, snail, snake, streetcar, sweet_pepper, tank, tiger, tulip, turtle, willow_tree, wolf, worm

