# OpenReview forum: "Few-shot Learning by Focusing on Differences"
_ICLR.cc/2020/Conference — Reject_

### Official Review · AnonReviewer3 · 2019-10-22
**Official Blind Review #3**

**Rating:** 3

**Review:**

In this paper the authors propose a metric based model for few-shot learning. The goal of the proposed technique is to incorporate a prior that highlight better the dissimilarity between closely related class prototype. Thus, the proposed paper is related to prototypical neural network (use of prototype to represent a class) but differ from it by using inner product scoring  as a similarity measure instead of the use of euclidean distance. There is also close similarity between the proposed method and matching network


overall, the paper does not  highlight the novelty of their proposed method especially prototypical network and matching network. Thus, the related work session is so general and does not tackle the close models in details. The experiments do not provide convincing evidence of the correctness of the proposed approach.

Several parts are unclear/incomprehensible:
(1) The Introduction is confusing and does not demonstrate the problem that the paper is trying to solve. Specifically, the described intuition (Mill’s method of difference) is not convincing
(2) the first sentence of  the section “Our work.”  (page 1) is long and unclear …  “In this paper, we propose a model that focuses on the differences in the support set of closely related classes in assigning the class label to a new instance in the novel task.”
(3) the use of the two level of embedding is confusing and not clear. Figure 1 is also confusion and not clear.

the correctness of the proposed approach is not proved by the conducted experiment and does not provide convincing and fair comparison with SoA techniques:
(1) The experiments do not provide the details of the used architecture compared to your baseline.  (how many layers are used in both embedding systems)
(2) In Table 1 you are using the results reported by Chen et al. (2019) did you use his framework (Resnet or 4 layers CNN)

**Minor comments**
The definition of the embedding function f = (f_g o f_f)  (in line 1 page 5) is not consistent with the domain of each function  f_g is defined on R^H x R^H.

K is not defined (last line page 3)


**Experience Assessment:**

I have read many papers in this area.

**Review Assessment: Checking Correctness Of Derivations And Theory:**

I assessed the sensibility of the derivations and theory.

**Review Assessment: Checking Correctness Of Experiments:**

I carefully checked the experiments.

**Review Assessment: Thoroughness In Paper Reading:**

I read the paper at least twice and used my best judgement in assessing the paper.

---

> ### Author Response · Authors · 2019-11-15
> **Clarification for Official Blind Review #3**
>
> Thanks for reviewing our paper.
>
>
> Similarity with Prototypical Networks and Matching Network
> ---
> Our proposed method is similar to the prototypical networks (Snell et al., 2017) -- and subsequently Mensink et al. (2013) -- in its use of mean representation of class (or prototypes). The similarity stops there, as the prototypical networks directly perform classification by comparing the distance of the new input to each prototype. They assume that the embedding function that produces the prototypes can sufficiently capture useful and general enough representation that is transferable to the novel set; it only computes global task embedding. As shown in our results, their assumption breaks down when there are changes in class granularity or the label is of fine granularity. In contrast, our method does not directly classify on the prototypes; instead, it transforms the prototypes, producing task-aware embeddings that are locally orthogonal to the shared components belonging to different classes. Thus, classification is performed by computing a Softmax over dot product between the new point and the task-aware prototypes.
>
> The Dissimilarity Network uses context embedding similar to the full context embedding (FCE) extension of the matching network (Vinyals et al., 2016). However, there are some glaring differences in how they operate. The matching network carries the entire support for the prediction. It predicts the label of an unknown point by computing the linear combination of the label of its support set; as the support set grows, the memory increases linearly with it. Their full context embedding conditions the prediction on the entire support set; computing the task-aware embedding is quadratic in the number of elements of the support set. Moreover, as they do not construct an explicit reference for the classes to condition into, it is less clear how reasonable separation of a point belonging to different classes can be maximized. Our model only maintains a set of prototypes and classifies a new point based on how orthogonal its representation to the prototypes. Since we condition the prediction only on the prototypes, it will not grow with the size of the support set. Moreover, computing the task-aware embedding is only quadratic in the number of prototypes (i.e., labels) -- as opposed to the number of support set. We also explicitly computes the representations to be dissimilar; lifting the reliance on learning sufficiently separable inter-class representations only to the loss function.
>
> We have also updated our related works to give more focus on the metric and similarity learning methods.
>
>
> Novelty, Mathematical Inaccuracies, & Clarity issues
> ---
> We have updated the paper to better highlight the novelty of our work, as well as incorporating the comments given.
>
>
> Comparison Fairness
> ---
> Four-layer convolution backbone (Conv-4) with an input size of 84x84 is used as a feature extractor for all methods for a fair comparison. Apart from that, our method is parameterized by the self-attention with 2 layers BLSTM for query, key, and value. The rest of the methods follow exactly from their description in their respective papers. For example, RelationNet is parameterized by an additional 2 layers of convolution networks and 2 fully-connected layers.

---

### Official Review · AnonReviewer1 · 2019-10-23
**Official Blind Review #1**

**Rating:** 3

**Review:**

The authors propose a new neural network model, called as Dissimilarity Network, to improve the few-shot learning accuracy.
Overall the idea is well motivated that by emphasizing the difference among classes, the model can achieve more accurate predictions for classes where only limited data points are available for training.
However, the paper is not quite well written.
Firstly, much of the work is built upon previous work including attention mechanisms, episodic training for few-shot learning. Such components are the core of this work because the attention mechanisms implement the class-awareness, and the episodic training facilitates the LSTM structure. Yet these are not well explained and not much context is provided, thus making the paper hard to follow.
Secondly, some terms are fairly overloaded, or not clearly defined. For example, the “prior” as mentioned in both the abstract and the introduction doesn’t refer to the commonly interpreted term as in the Bayesian settings, but rather as a hand-waiving term to indicate the model design. Also, the terms, “score”, “metric”, “dissimilarity” are mentioned in the paper but the paper is not really learning the metric, to my understanding. Thus the details of the paper is quite hard to grasp.
Lastly, the idea of designing the global embedding and the task aware embedding is interesting but shouldn’t really be restricted to few-shot learning. It would be interesting to test the idea on general classification tasks, for example in a simple cross validation settings.
Thus I think the paper would be stronger if the above are addressed and it’s not ready for publishing yet in its current form.

Below are some more detailed comments:
1)	In the abstract, the “newly introduced dataset H-CIFAR” is not precise to me; my understanding is that the paper proposes such an experiment design for testing how well a classifier can predict the labels with hierarchy. The current writing refers to that the authors comprises a completely new dataset with new labels.
2)	In the last sentence of the second paragraph in Introduction, the question is asked “what prior” should be reasonable. Since the authors didn’t really add any priors in a Bayesian settings but rather designed an architecture, I suggest to reword something like “how to explicitly encode hierarchies into the model structure”.
3)	In Section 2.1, some more description for “episodic training” would be nice: why should it be used? How is it used and why it makes sense in the few-shot learning context?
4)	In Section 2.2, it would be nice to add the mathematical definition of “prototype”.
5)	In Section 2.2.1, it would be nice to define “H”.
6)	In Section 2.2.2, is M required to be fixed given it’s episodic training? Also it would be nice to add more details about the attention mechanism.
7)	In the result section, it would be nice to discuss when the proposed method is doing better than other methods, for example RelationNet, as well as when it’s worse since different datasets show different results.


**Experience Assessment:**

I have read many papers in this area.

**Review Assessment: Checking Correctness Of Derivations And Theory:**

I assessed the sensibility of the derivations and theory.

**Review Assessment: Checking Correctness Of Experiments:**

I assessed the sensibility of the experiments.

**Review Assessment: Thoroughness In Paper Reading:**

I read the paper at least twice and used my best judgement in assessing the paper.

---

> ### Author Response · Authors · 2019-11-15
> **Clarification for Official Blind Review #1**
>
> Thanks for taking the time to review our paper.
>
>
> Explanation about Components
> ---
> We have updated the manuscript to include more explanation about the methods that we rely upon.
>
>
> Terms & Similarity Learning
> ---
> We have updated the manuscript to use more appropriate terms: inductive bias and similarity learning.
>
> Our method adopts the approach of similarity learning. Instead of learning a distance or similarity function, we learn a space (embedding) that works well with a fixed similarity-based classifier in that space. Specifically, our model learns to construct a space that is optimized to separate data that belong to different classes for a classifier that uses dot-product as its similarity function.
>
>
> Application on General Classification Tasks
> ---
> In the usual classification tasks, the labels are fixed during the training and testing. Our approach is advantageous if the labels are changing between task -- as in few-shot classification.
>
>
> Detailed Comments
> ---
> We have updated our manuscript based on the above comments. We also updated the name of our harder variant of CIFAR dataset to CIFAR-Hard to avoid confusion -- as CIFAR is already hierarchically labeled. For point (1), it’s true that our new dataset can be seen through the lens of experiment design. However, it’s also true that the dataset is new, in a sense that the dataset comprises of a new set of image label pairs, with the label derived from the original CIFAR.
>
>
> Results & Discussions
> ---
> We have added more discussion about the results in the manuscript. Please have a look at the updated manuscript.

---

### Official Review · AnonReviewer2 · 2019-10-24
**Official Blind Review #2**

**Rating:** 3

**Review:**

The stated contributions of the paper are: (1) a method for performing few-shot learning and (2) an approach for building harder few-shot learning datasets from existing datasets. The authors describe a model for creating a task-aware embedding for different novel sets (for different image classification settings) using a nonlinear self-attention-like mechanism applied to the centroid of the global embeddings for each class. The resulting embeddings are used per class with an additional attention layer applied on the embeddings from the other classes to identify closely-related classes and consider the part of the embedding orthogonal to the attention-weighted-average of these closely-related classes. They compare the accuracy of their model vs others in the 1-shot and 5-shot setting on various datasets, including a derived dataset from CIFAR which they call Hierarchical-CIFAR.

Overall, while we like the concepts/ideas and the problem is definitely important, we were not enthusiastic about the paper. First, we found the write up to be cryptic, involving very long unclear statements. It read as if the authors were writing for themselves and not for ICLR general audience. Beyond the writing style, we found the paper to have:

* Inadequate description of the model, including mathematical inaccuracies.
* Inadequate description of Hierarchical-CIFAR, motivation, and evaluation.

Description of model:
------------------------------
The manuscripts describe the presented approach as metric learning but make no use of a distance function in the different spaces they map to. Instead, the manuscript defines an inner product over embeddings to compute similarities.

The manuscript describe their “self-attention operation” as a dynamic set-to-set operation. While the usual definition of self-attention is permutation invariant, the definition presented here is not and thus cannot be accurately described as a mapping between sets. Specifically, in the usual presentation of self-attention the only sharing of information between different set elements is during the outer product of the key and query vectors. The use of a BLSTM between “neighboring elements” of the set of prototype vectors violates this assumption and induces a lack of permutation invariance. This makes the method sensitive to permutations of classes, which does not make sense for predicting unordered classes.

In sections 2.2.3 and 2.2.4 the material is presented twice but slightly differently. For example, the definition of $b_k$ inline before equation 3 differs from equation 6 later in the text.

Equation 7 is incorrect and should not exclude the current class from the denominator.

The description of how to classify new points after equation 6 is poorly explained. The description of what happens when $h_V$ is a “BLTSM” [sic (should be BLSTM)] is noninformative.

The manuscript describes two dimension sizes $H$ and $M$ but the definition of $attn$ requires that $H = M$.

Description of new dataset and evaluations:
------------------------------------------------------------
One of the stated contributions of the manuscript is a methodology to build harder few-shot learning datasets. Section 4.2 is the only place in the text that appears to address this point, but is unclear where either new finer-grained or coarser-grained labels are coming from (new manual annotation or otherwise). The manuscript “leave[s] out the detail of its construction for simplicity”, but it is unclear what is being done here in the first place.

The manuscript does not detail tuning competing methods on the new dataset and so it is unclear whether it is a fair comparison.

The manuscript presents evaluations without any discussion of differences in performance between datasets or the 1-shot/5-shot settings. For example, their method is significantly better on CUB on 1-shot but not so much on 5-shot, on the other hand it is not significantly better on 1-shot for H-CIFAR and CIFAR-HS but then becomes better than the rest with 5-shot.

Additional comments/corrections
---------------------------------------------
There were numerous typos and grammatical errors that were present in the manuscript that did not directly impact this evaluation but should be fixed in the future.

**Experience Assessment:**

I have read many papers in this area.

**Review Assessment: Checking Correctness Of Derivations And Theory:**

I assessed the sensibility of the derivations and theory.

**Review Assessment: Checking Correctness Of Experiments:**

I assessed the sensibility of the experiments.

**Review Assessment: Thoroughness In Paper Reading:**

I read the paper at least twice and used my best judgement in assessing the paper.

---

> ### Author Response · Authors · 2019-11-15
> **Clarification for Official Blind Review #2**
>
> Thank you very much for your review.
>
>
> Similarity Learning
> ---
> We have updated our manuscript and present the approach as similarity learning. We learn a function that maps to a space that is optimized for dissimilarity. Since our notion of dissimilarity is based on vector orthogonality, it’s only natural that we use an inner product as a similarity function in such space.
>
>
> Permutation Invariance
> ---
> We are aware of the sequential nature of BLSTM, which can be counter-intuitive as we are modeling a set-to-set operation which shouldn’t have any preference for ordering. However, we found empirically that this setup offers more performance gain compared to the use of traditional linear function as attention embedding function. The BLSTM may learn to ignore the unimportance of set ordering due to the nature of episodic training, which exposes it to many permutations of the possible class-orderings. Moreover, the attention also gives a global context of the member of the set, which could further alleviate the ordering issues (if any).
>
>
> Harder Benchmark
> ---
> Our approach requires the dataset that we derived from to have at least two different levels of class granularity. For example, CIFAR dataset which has two levels of labels granularity. ImageNet labels also form a hierarchy which -- through this method -- can be derived into several hard few-shot classification datasets. In the case where different labels of granularity are absent, one may be able to construct new labels by exploring the natural hierarchy which may present.
>
> We leave out the detail of the construction of the validation set, not the dataset.
>
>
> Comparison Fairness
> ---
> No finetuning is performed on CIFAR-H and CIFAR-FS, all the hyperparameters are the one used in the mini-ImageNet dataset. This applies to all methods (including ours).
>
>
> Results & Discussions
> ---
> We have added more discussion about the results in the manuscript. Please have a look at the updated manuscript.
>
>
> Mathematical Inaccuracies, Clarity, and Grammatical Errors
> ---
> We have updated the manuscript to address the issues.

---

### Decision · Program_Chairs · 2019-12-19

**Decision:**

Reject

**Comment:**

Main content:

[Blind review #3] The authors propose a metric based model for few-shot learning. The goal of the proposed technique is to incorporate a prior that highlight better the dissimilarity between closely related class prototype. Thus, the proposed paper is related to prototypical neural network (use of prototype to represent a class) but differ from it by using inner product scoring  as a similarity measure instead of the use of euclidean distance. There is also close similarity between the proposed method and matching network.

[Blind review #2] The stated contributions of the paper are: (1) a method for performing few-shot learning and (2) an approach for building harder few-shot learning datasets from existing datasets. The authors describe a model for creating a task-aware embedding for different novel sets (for different image classification settings) using a nonlinear self-attention-like mechanism applied to the centroid of the global embeddings for each class. The resulting embeddings are used per class with an additional attention layer applied on the embeddings from the other classes to identify closely-related classes and consider the part of the embedding orthogonal to the attention-weighted-average of these closely-related classes. They compare the accuracy of their model vs others in the 1-shot and 5-shot setting on various datasets, including a derived dataset from CIFAR which they call Hierarchical-CIFAR.

--

Discussion:

All reviews agree on a weak reject.

--

Recommendation and justification:

While the ideas appear to be on a good track, the paper itself is poorly written - as one review put it, more like notes to themselves, rather than a well-written document to the ICLR audience.